# Evaluating Teaching Effectiveness of Medical Humanities in an Integrated Clerkship Program by a Novel Prospective Propensity Score Matching Framework

**DOI:** 10.3390/ijerph19031882

**Published:** 2022-02-08

**Authors:** Chen-Huan Chen, Shuu-Jiun Wang, Wan-Yu Yeh, Chung-Li Wu, Yong A. Wang, Cheng-Feng Chen, Ying-Ying Yang, William J. Huang, Kwan-Yee Chan, Chi-Wan Lai, Ging-Long Wang, Hao-Min Cheng

**Affiliations:** 1School of Medicine, National Yang Ming Chiao Tung University, Taipei 11217, Taiwan; chchen3@nycu.edu.tw (C.-H.C.); sjwang@vghtpe.gov.tw (S.-J.W.); yangyy@vghtpe.gov.tw (Y.-Y.Y.); jshuang@vghtpe.gov.tw (W.J.H.); 2Department of Medical Education, Taipei Veterans General Hospital, Taipei 11217, Taiwan; wanyu.research@gmail.com (W.-Y.Y.); chungliwu2013@gmail.com (C.-L.W.); 3Department of Neurology, Taipei Veterans General Hospital, Taipei 11217, Taiwan; 4Center for Evidence-Based Medicine, Taipei Veterans General Hospital, Taipei 11217, Taiwan; 5Koo Foundation Sun Yat-Sen Cancer Center, Taipei 11217, Taiwan; wang@kfsyscc.org (Y.A.W.); sammy@kfsyscc.org (C.-F.C.); kychan@kfsyscc.org (K.-Y.C.); cwlai@kfsyscc.org (C.-W.L.); glw@kfsyscc.org (G.-L.W.); 6Department of Urology, Taipei Veterans General Hospital, Taipei 11217, Taiwan; 7Program of Interdisciplinary Medicine (PIM), National Yang Ming Chiao Tung University College of Medicine, Taipei 11217, Taiwan; 8Institute of Public Health and Community Medicine Research Center, National Yang Ming Chiao Tung University College of Medicine, Taipei 11217, Taiwan

**Keywords:** effectiveness, clinical training, propensity modelling

## Abstract

Background: This study aims to rigorously compare the effectiveness of the educational programs of a new integrated clinical clerkship in medicine (3 months) and surgery (3 months) at a cancer center with the conventional subspecialty-based rotations at a tertiary teaching hospital, by this prospective, pre-post comparative method. Methods: Between 2013–2016, we compared 69 students who had selected the integrated clerkship that emphasized clinical competency and medical humanities training with 138 matched peers who had completed conventional clerkships during the same period. Outcome measures for medical humanities included empathy, patient-centeredness, and other values and skills related to holistic health care professionalism by introducing prospective propensity score matching (PSM). Results: At baseline, no significant between-group differences existed. At the completion of the core clerkships, students receiving the integrative clerkship had significantly higher scores on the Patient–Practitioner Orientation Scale (PPOS) and the Professionalism Climate in Clinical Teaching Environment (PCI), and similar Jefferson Scale of Physician Empathy Student Version (JSPE) scores, as compared with the comparison group. We also found that the students in this program did not perform worse than those in the traditional internship group in the comprehensive and formative OSCE medical clinical skills test. Conclusions: Our study develops an empirical basis for rigorous evaluation to design medical education to improve the medical humanities values and skills of interns. Features of the new integrated clerkship program that we developed include substantial participation by the students in patient-centered in-hospital culture, as well as reflection, discussion, and feedback on actual clinical cases.

## 1. Introduction

To cultivate an informed, curious, compassionate, proficient, and moral physician who can serve the needs of future societies [1], the models of clinical training require continuous improvement and innovation in the era of explosive progress and demand in medical care [2,3]. Rigorous evidence is required to support new models, such as interdisciplinary and continuity-of-care clerkships [4,5,6,7,8,9], district hospital-based clerkships [10,11,12]; and nontraditional clinical clerkships [13] are truly better than the conventional ones. Because it is very difficult to evaluate a new model of clinical clerkships using the experimental design of randomized control trial [14], most studies report quasi-experimental results, namely, the direct comparison of student performance between those who voluntarily received the new clerkships and part or all of those who received the conventional clerkships, with post-hoc adjustments for the differences of the baseline characteristics between the two non-randomized groups [4,5]. The use of a convenient but unselected comparison group in the study design would likely introduce selection bias that may dilute or overstate the strength and weakness of the new clerkships and lead to inappropriate decision making, which has also become an obvious unmet need of medical education.

In Taiwan, as well as in many western countries, clinical clerkships are usually conducted in tertiary general hospitals affiliated with the medical schools, where patient care is highly specialized and efficient with fast inpatient turnover, so the novice medical students have very limited time, opportunities, or supervision for hands-on experience in developing their empathy and skills in patient–doctor relationships, in addition to the core clinical skills such as performing history taking and physical examination. Therefore, we designed and evaluated a novel clinical clerkship at a cancer center that focuses on humanities education and building clinical skills through direct patient care responsibilities, with long-term direct supervision by a teaching group of trained clinical teachers and designated mentorship by senior physicians [15]. Our preliminary report of the first 3 years (2006–2007, 2007–2008, 2008–2009) results suggested that the pilot collaborative program was a successful model for clinical education in the teaching of core clinical competencies through direct patient care responsibilities, that both emphasizes essential clinical skills and provision of holistic care [15].

The promotion of holistic medical care is a focus of Taiwan’s medical education reform in recent years [16], and it also responds to the international movement to address the importance of the holistic medical environment [17]. In Taiwan, medical students are usually the top academic performers in high school, and the medical education system also puts high emphasis on medical professional knowledge and skills [18]. With the introduction of the concept of holistic medical education, “integration of human multi-faceted intelligence and life experience” has been considered to be a very important direction for developing a doctor with both medical technology and humanistic care [17,18]. However, when we reviewed previous studies which explored the integration of holistic medical education into clinical practice, research on students’ performance in professional skills and humanistic care was rare. Preliminary research has shown that the acceptance of holistic care of medical students is related to their learning motivation, ethical cognition, and even mental health [18,19,20]. We believe that empirical evidence that evaluates the teaching effectiveness of medical humanities in an integrated clerkship program in medical education has considerable importance for catalyzing the holistic education paradigm.

In the present study, we prospectively enrolled a comparison group of medical students who selected to receive conventional clerkships in medicine and surgery and had baseline characteristics matched to that of participants who selected to receive integrated clerkships, in order to rigorously evaluate the humanities education integrated in the clinical clerkship in medicine and surgery in a clinical setting very different from a tertiary medical center. For the first time, we applied a propensity modelling scheme to calculate a propensity score for every student in the class according to the prespecified personal characteristics and personality dimensions, and then identified and invited two best matched comparison students for each participant of the integrated clerkships.

The main purposes of this study were: (1) to evaluate the training effect of the Integrated Clerkship Program on the medical humanistic literacy of trainee medical students; (2) to understand the comparative effectiveness between the Integrated Clerkship Program implemented in the specialized hospital and the conventional internship program of the tertiary teaching hospital, and to identify whether there are differences in clinical skill performance among participating medical students.

There were two corresponding hypotheses of the present prospective comparison study of educational interventions: First, students receiving clinical clerkship training in an integrated common learning module of medical humanities may outperform their matched counterparts in humanity outcome assessment; second, these students in this integrated clerkship program may have clinical performance non-inferior to the matched students receiving conventional clinical training at a tertiary medical center.

## 2. Materials and Methods

### 2.1. The Integrated Clinical Clerkship Program

The design and implementation of the integrated clinical clerkship in medicine and surgery has been detailed previously [15]. In brief, the A University, one of the three government-funded medical schools in Taiwan, implemented a new 7-year problem-based learning curriculum in 2002 that included 9-month core clerkships in the fifth year and 22-month internships in the remaining sixth and seventh years. The core clerkships consisted of three blocks of rotations in medicine (3 months), surgery (3 months), and a combination of obstetrics/gynecology, pediatrics, and radiology (3 months). The clerkships were conducted at B Hospital, a tertiary medical center. After completion of the clerkships, each medical student received internships at B Hospital or other branches located in central or south Taiwan; all three are tertiary teaching hospitals affiliated to A University. Since year 2006, the C Hospital—a 300-bed comprehensive cancer center whose mission is to provide state-of-the-art holistic cancer care and to promote excellence in medical education—collaborated with A University to implement the novel integrated clerkship rotations lasting 3 months each in general medicine and surgery. A maximum of 18 students among a class of around 130 fifth-year medical students could select the integrated clerkship program after interview by C Hospital faculty, to substitute the conventional rotations in medicine (3 months), surgery (3 months) at B Hospital.

The details of the integrated clerkships at C Hospital have been published elsewhere [15]. Although C Hospital is a cancer center, two general medicine teaching teams, each consisting of an attending physician, a resident, interns, three clerks, and a nurse care manager were set up to provide general medical service on a dedicated general ward with a designated nursing unit for the program. Each student took care of one to three patients concurrently and continuously from admission to discharge and switched to another teaching team on the same ward to receive progressive patient care responsibility in both quantity and quality across the three months of the general medicine clerkship. Through the direct responsibility for patient care and self-directed learning, students acquired all essential competencies including history taking, physical examination, patient presentation, medical record writing, interpretation of imaging and laboratory tests, clinical reasoning, communication with patients/families and colleagues, and professionalism. The surgery clerkship comprised three monthly rotations in three subspecialties and resembled the general medicine clerkship in its structure, administrative support, and focus on basic clinical skills. The surgical clinical teams featured two attending physicians to increase variety in cases and teaching styles, and students’ participation in outpatient clinic to learn indications and contraindications for surgery, informed consent, clinical assessment, and workup in patients with an unestablished diagnosis.

Because the clinical care for cancer patients is increasingly complex with numerous and sophisticated therapies, students must learn to be sensitive to patients and their families’ needs and issues of patient safety, quality improvement, medical delivery systems, and doctor–patient relationship. Therefore, medical humanities were integrated as a common learning module continuing throughout rotations in medicine and surgery at C Hospital to facilitate the building of ethics, empathy, and patient–doctor relationship, and professionalism [15]. Students discussed patient and physician perspectives on life, death, and medical care, the humanity issues encountered in the daily care of their patients in the fortnightly humanities-in-medicine seminar moderated by psychiatrists and senior clinicians. Cases, topics, or questions were submitted by students beforehand so that relevant reading materials or references could be identified by the session moderators to help facilitate the discussion. Clinicians and social workers involved in the patient’s care were invited to attend to enrich the discussions with different perspectives. Moreover, students were encouraged to select an optional module to follow a cancer patient over the entire clerkship period. 

### 2.2. The Conventional Clerkships in Medicine and Surgery

B Hospital is the principal teaching hospital of A University and receives medical students from A University and other medical schools in Taiwan for clerkship and internship trainings. Directors of the clerkship program at B Hospital are appointed by A University and are responsible for the design, implementation, and monitoring of the clerkship rotations. Each student selected three out of ten subspecialties in medicine for monthly rotations and five (general surgery I, general surgery II, colorectal surgery, pediatric surgery, and chest surgery) plus one out of another six subspecialties in surgery for half-monthly rotations. Teaching teams in each subspecialty consisted of one to two attending physicians, a resident, and one to two clerks, with or without an intern. No formal medical humanities courses or training were provided during the clerkship or internship trainings. 

In both arms, emphasis was also placed on student support throughout the clerkship period. Each student was assigned a mentor, an experienced physician not involved in the grading of the student, to provide ongoing guidance and support. In addition, monthly one-on-one feedback was provided by the clerkship directors. Through the dedicated clerkship administrator and team care managers, the students had additional opportunities to seek support and feedback. 

Before participating in the internship, both groups of B Hospital and C Hospital students had taken medical ethics courses, and the basis for the comparison of learning effectiveness between the two groups was comparable.

### 2.3. Enrollment of Participants and Sample Size Calculation

The present study was planned to enroll all 18 fifth-year medical students who selected the C Hospital integrated clinical clerkship and twice the number of their peers who selected the B Hospital conventional clerkship for the academic years of 2013–2014, 2014–2015, 2015–2016, and 2016–2017 (Figure 1). To ensure successful enrollment, all fifth-year medical students were invited to complete four questionnaires, namely, International English Big-Five Mini-Markers (Big-Five) [21,22], Task of Medicine Scale (TOMS) [4,23], Jefferson Scale of Physician Empathy Student Version (JSPE) [24], and Patient–Practitioner Orientation Scale (PPOS) [25,26], after the briefing of the research project to the whole class and before the selection of the clerkship programs by the students. In order to maximize the number of participants, the researchers endeavored to invite all medical intern students from Hospital C during the research period. Through the method of prospective propensity score matching, students with similar basic background conditions as the students of Hospital C were selected to prospectively compare the teaching effectiveness of the integrated clinical clerkship program. Based on personal characteristics including age, gender, admission route, student loan, part-time job, academic achievement in the preclinical years, and personality dimension scores from the Big-Five, a propensity score was generated for each C Hospital student; and then two B Hospital peers with a best-matched propensity score were invited to join the study (36 students per academic year). The research project was approved by the institutional review committees at B Hospital and C Hospital. Written informed consent was obtained from all students in the class before they started to fill in the questionnaires. 

### 2.4. Study Design

To study both the maturation effects and final outcomes [27], students were invited to complete the questionnaires for ethics, empathy, and patient–doctor relationship at three time points. Before starting the clerkships, all participating students had already completed four questionnaires: Big-Five, TOMS, JSPE, and PPOS (Figure 1). Repeat TOMS, JSPE, and PPOS questionnaires were taken after completion of the first 3-month clerkship in medicine or surgery (Figure 1). At the end of the 9-month core clerkships, all participants repeated TOMS, JSPE, and PPOS, and took the additional Professionalism Climate in Clinical Teaching Environment (PCI) questionnaire (Figure 1) [28].

### 2.5. Self-Report Instruments

The 40-item Big-Five set has been validated for cross-cultural applications [21]. The validated Chinese version includes 40 items on nine-point scale to provide scores on five personality dimensions of extraversion, agreeableness, conscientiousness, neuroticism, and openness [22]. TOMS includes a list of eight tasks or goals that participants will be asked to rank by importance [4,23]. Four are physiological (biomedical), such as “to conduct a thorough physical exam” and four involve mental sociality (psychosocial), for example, “to understand the patient’s perspective about the problem”. JSPE has been widely used to measure both cognitive and affective empathy in medical education [25]. The 20 items on a seven-point scale form three underlying factors: Perspective Taking, Compassionate Care, and Standing in the Patient’s Shoes. PPOS is a widely used questionnaire to measure the relationship between doctors and patients [25,26]. PPOS has 18 items on seven-point scale in two dimensions: a 9-item Caring subscale and a 9-item Sharing subscale. PCI contains 12 items on five-point scale about the frequency of professional and unprofessional behaviors observed in the clinical environment [28]. Students answered the same 12 items about their perceptions on student peers, residents, attending physicians, and faculty. All questionnaires were checked to evaluate if they had been completed before they were collected, and the questionnaires were analyzed only when the information was complete.

### 2.6. Objective Assessment of Clinical Performance

A University incorporated a summative Objective Structured Clinical Examination (OSCE) designed and conducted by the faculty of B Hospital at the end of the block rotations in medicine, surgery, and a combination of obstetrics/gynecology, pediatrics, and radiology. Students participating in the C Hospital integrated clerkships in medicine and surgery joined the same OSCEs with their B Hospital peers.

### 2.7. Clinical Grades

At the end of each monthly or half-monthly rotation during the clerkship and internship training at both hospitals, the supervising attending physician is responsible to assess the student’s overall clinical performance and give a summative grade (range, 0–100) for the rotation. 

The assessment and analysis of student performance was performed by outcome assessors and statisticians blinded to the allocation, respectively. OSCE scores and internship scores were measured before the end of the school year, and the objective scoring standards had been established before conducting all of the scoring procedures. The teacher responsible for the assessments had been reminded that the allocation should not affect the scoring during the research.

### 2.8. Evaluation of Humanities Education

The humanities education at C Hospital included the designated medical humanities learning module integrated in the clerkships in general medicine and surgery, and hidden curriculum in direct patient care. We used TOMS, JSPE, and PPOS as instruments to measure the changes in ethics, empathy, and patient–doctor relationship over time, and PCI at the end of 9-month core clerkships to compare the professionalism of the environment between the two hospitals.

### 2.9. Statistics Analysis

Categorical variables were expressed as frequencies and percentages. Continuous variables were summarized as means and standard deviations. Chi-square test and Student’s *t* test were used to test for categorical and continuous variables between the two hospitals, respectively. To simulate the nature of the randomization, we performed the propensity score matching to minimize the impact of imbalances of baseline characteristics. To generate a propensity score, we constructed a logistic model to predict the probabilities of allocation to the C Hospital. Variables included in the propensity score modelling were age, gender, admission route, student loan, academic achievement in the preclinical years, and personality dimension scores from the Big-Five. Based on the calculated propensity score, a 2:1 matching method, without replacement, was used with the method of nearest neighbor matching. We assessed the success of propensity score matching by comparing baseline characteristics between C Hospital and B Hospital using Student’s *t* test. Given that the ranges of the original scores of the assessment tools used in this study varied, all scores were linearly transformed to a 0–100 scale to facilitate the analysis, comparison and interpretation of the assessment. Non-inferiority tests have been adopted in other medical education studies in recent years [24,25] and were used to assess if the learning outcomes of those who received clinical training in C Hospital were not superior to those who received clinical training in B Hospital. Non-inferiority margins of each learning outcome were derived by calculating the difference between the lower limit of the 1st and 2nd quartile within the 1st tertile of the whole class in the academic year of 2013. Generalized estimating equations were used to account for the correlated structure of our study data. Specifically, we compared the learning outcomes of B Hospital and C Hospital from three time points using generalized estimating equations with the identity link function, and an unstructured working correlation matrix.

After the first-year program, a sample size of 50 and 97 subjects in the integrated and conventional core clerkship programs were estimated (effect size of PPOS sharing, 6.75; standard deviation, 9; alpha, 0.003; beta, 0.2; proportion of the integrated program, 0.333). Therefore, a prospective enrollment for a consecutive four-year period was determined on the commencement of the second year of the study.

In addition, the baseline effects were accounted for in all statistical models when estimating the overall group effect of the humanities outcome between the two hospitals. Statistical analyses were conducted with SAS 9.4 [29]. Propensity score matching was performed using MatchIt version 3.0.2 in R 3.4.1. With Bonferroni correction for the multiple testing issue for the comparisons of humanities outcomes between the two hospitals at the end of the clinical program, a significance level at 0.033 was considered, and all other statistical analyses were considered at a significance level of 0.05.

## 3. Results

A total of 69 C Hospital and 138 B Hospital clerks participated in the study and completed all required questionnaires and OSCEs (Figure 1). Their baseline characteristics are shown in Table 1. The two groups had similar personal characteristics including age, sex, admission route, study load, part-time job, and personality dimensions. They also had similar scores in TOMS, JSPE, and PPOS before starting the clerkships.

TOMS, JSPE, and PPOS were measured at 3 points in time. The average TOMS psychosocial score of B Hospital students fell during follow-up and remained low at the end of the core clerkship (Figure 2). In contrast, the C Hospital students showed a slight fall during follow-up but rose to the level of baseline so at the end of the 9-month core clerkships, the C Hospital students had a higher TOMS psychosocial score than their peers (Figure 2, Table 2). Over the 3 time-points, the C Hospital students had a significantly higher TOMS psychosocial score than their peers (Figure 2; p for overall group effect = 0.020). In contrast, the average TOMS biomedical score of B Hospital students increased more sharply than that of C Hospital students and the differences remained significant at the end of the follow-up (Appendix A).

The JSPE Perspective taking score in the C Hospital students remained unchanged across the three time points (Figure 3). In contrast, the score in the B Hospital students fell during follow-up and was significantly lower than their peers, and then the score rose to close to the level of baseline at the end of the core clerkships. Over the three time points, the C Hospital students had a significantly higher JSPE Perspective taking score than their peers (Figure 3A; *p* for overall group effect = 0.008). Results were similar for the JSPE Compassionate Care score (Figure 3B; *p* for overall group effect = 0.024) and JSPE total score (Figure 3D; p for overall group effect = 0.021).

However, the JSPE Standing in patient’s shoes score significantly increased with time in both groups (*p* for trend: 0.0092 in B Hospital and 0.0004 in C Hospital), but the slope appeared steeper in the C Hospital students (Figure 3C; *p* for group-time interaction term = 0.219). However, there was no significant between-group difference at any time-point (Figure 3C; Table 2).

The PPOS sharing score increased slightly during follow-up and remained higher than baseline to the end of clerkship in the C Hospital students (Figure 4A). In contrast, the score significantly decreased with time in the B Hospital students (*p* for trend < 0.0001) so the C Hospital students had a significantly higher score than their peers during follow-up and at the end of the clerkship (Figure 4A; p for group-time interaction term = 0.036, *p* for overall group effect <0.0001; Table 2). Results were similar but less significant for the PPOS caring score (Figure 4B; p for overall group effect = 0.029). The C Hospital students had a significantly higher PPOS caring score than their peers at the end of the clerkships only (Figure 4B; Table 2).

PCI was taken only at the end of the clerkships. The C Hospital students had higher PCI score to peers (*p* = 0.0490), and a significantly higher PCI scores to peers (*p* = 0.0490), residents (<0.0001), attending physicians (*p* < 0.0001), and faculty (*p* < 0.0001).

Both groups had similar OSCE scores in the blocks of medicine, surgery, and a combination of obstetrics/gynecology, pediatrics, and radiology. Both groups also had similar average grades for the sixth and seventh years of the internship (Table 2).

## 4. Discussion

### 4.1. Main Findings

In the present interventional study for clinical education, a wide variety of comparative learning outcomes between two clerkship programs were serially evaluated by using a rigorous assessment framework, which prospectively matched students from the integrated and conventional clerkship programs with propensity score modeling before the start of the clinical training. As compared with students receiving the conventional clerkships in medicine and surgery in a tertiary teaching hospital, students receiving the integrated clinical clerkships in a cancer center had superior humanity outcomes assessed by TOMS, JSPE, PPOS, and PCI, with less significant differences in JSPE scores. Moreover, their objective clinical achievements including OSCE and clinical grades of the sixth and seventh years of the internship were non-inferior to their matched counterparts.

Medical schools in Taiwan have been introducing humanities into medical education for decades [18], make tremendous efforts to continually deepen the “holistic medical education”. Several practical strategies have been proposed to enhance curricula related to the humanistic aspects of medical training for medical students, including positive role modeling, establishing a humanistic learning environment, creating learning objectives directly related to psychosocial issues, and service learning [18,30,31,32]. These approaches to holistic education cover a wide range of philosophical orientations and pedagogical practices [18]. Broadly speaking, holistic education includes intelligent and professional clinical technology, as well as the improvement of spiritual aspects such as ethics, caring, and spirituality [18,20]. However, although these methods are theoretically beneficial to the cultivation of the humanism of medical students, the effectiveness of these methods or curricula have seldom been rigorously examined. In other words, there has been scarce evidence in enhancing the humanistic aspects of medical students. Our study represents one of the first efforts to produce evidence to inform education in the medical and health professions [18].

Instructing humanities and professionalism are longstanding challenges in a busy clinical environment, made even more difficult for the teaching faculty in medical centers who are pursuing academic excellence and clinical performance at the same time [33,34]. Medical students in their clerkship years at teaching hospitals develop professionalism mainly from observing their role models in the learning environment, without formal teaching in humanities [35]. However, in the past, specialized hospitals were often influenced by stereotypes and were considered to be less resourced to deliver holistic education during clinical practice, or less able to balance technical knowledge and non-technical literacy. In this study, we observed that at C Hospital, well-trained teaching faculty with a deliberately reduced clinical service load implemented a formal medical humanities course in a patient-centered, holistic care environment that was created by committed staff from many different disciplines in the hospital. Medical students were encouraged to discuss the details of biopsychosocial aspects of the care of cancer patients with their mentors and during scheduled weekly meetings [15]. The effectiveness of humanities learning outcomes demonstrated in the present study was apparently associated with the innovative design of the integrated clerkship in which formal humanities education is emphasized, and also related to the hidden curriculum in which role modeling may play an imperative role. Proper design of the clinical education program and the teaching environment overcomes the limitations of institutional resources and is conducive to the cultivation of the medical humanity of medical trainees. This speculation was supported by the findings of PCI (Table 2), through which the medical students rated the professionalism of their peers and teachers differently between the two hospitals.

There are several key contributors to the success of the new integrated clerkship program of A University in the enhancement of the humanistic aspects of medical students. First, during the routine seminars of case discussion held fortnightly, discussions and feedback surrounding medical humanities was provided regarding the care of their cancer patients. It has been suggested that the formative and summative feedback in medical education and professionalism learning is very useful in the development of students’ competency [36,37]. Secondly, the patient encounter in the care for medically and emotionally-needy cancer patients is unique, providing many opportunities for students to learn what to tell (or not to tell) patients, the challenges of communicating bad news, dealing with difficult families, coping with loss, and their own personal stresses. The skills of building up patient–doctor relationships and empathy are indispensable in the cancer center and therefore helpful for the development of professionalism of medical students. These factors contribute to the spirit of holistic education that focuses on the diverse life experiences of human experience, rather than being limited to basic technical training [36,37]. These may be the reasons that the students at the cancer center had a significantly higher rating of PPOS (sharing), and PCI at the end of their clerkship training (Table 2). As such, the program at the cancer center serves as a good reference for integrating humanities and professionalism education in the clerkships training for medical students [35]. The results of this study reflect the findings of past research that holistic care education can help improve medical student humanistic thinking [18,20], and present different aspects of medical humanity with more diverse and detailed indicators. In addition, the results of this study also support the feasibility of implementing an educational program in specialized hospitals that balances professional knowledge and humanistic care.

It must be noted that in the two participating hospitals, the training intensity and performance evaluation standards of students participating in clinical practice were comparable. According to the teaching hospital, accreditation evaluation provisions implemented by the Ministry of Health and Welfare (MOHW) in Taiwan [38], the number of clinical practice caring inpatients for medical students in the fifth grade starts with one and the upper limit is ten beds; the actual number of caring beds in Hospital B and C are both three or less.

To evaluate learning outcomes, randomized controlled trials are still the best design to study the effect of interventions. However, randomized controlled trials are not always feasible for clinical educational programs in the usual clinical setting. We therefore designed an assessment framework by using the prospective propensity score matching modeling to investigate the effectiveness of two different educational programs. As a surrogate for a randomized control trial, some possible confounding characteristics, including age, sex, admission route, having study loan, taking part-time job, and personalities of students, were evenly matched between groups by using the prospective propensity score matching modeling. In the present study, the matching effect of the propensity score matching was satisfactory, which was evidenced by the similar baseline characteristics as well as the baseline humanity assessment scores between two student cohorts.

In addition, the number of participants in this study was fulfilled because the sample number reached the pre-planning standard of the research plan.

### 4.2. Study Limitations

There are several limitations to the present study, which compared the learning outcomes between two educational programs. Although we endeavored to avoid the allocation or selection bias which inevitably make the subsequent between-group comparisons invalid, some unmeasured confounders relating to both the intervention assignment and learning outcomes might still exist. However, in view of the small differences of baseline measurements between two groups, the likelihood of the existence of such confounders with a strong impact on the differences of achieved questionnaire scores as well as other learning outcomes was small. In addition to the integrated clerkship program itself, the organizational structure and atmosphere of the two groups of hospitals may also have substantial impacts on students’ learning performance, limiting the comparability between these two hospitals. PSM can only control potential measured variables at the individual level.

On the other hand, due to the particularity of the level and type of hospitals, the medical humanities education environment of the setting and participant characteristics such as race, learning history, etc. in this study, whether the effect size of the teaching effectiveness in the Integrated Clerkship Program in this study can be generalized to other fields of medical education requires more in-depth studies and continuous verification.

In addition, the intervention for the two student cohorts was one year and the follow-up duration was only within one year. Whether such observed differences could be sustainable and endurable when these students are exposed to subsequent clinical training programs should be further studied. Professionalism is a complex, multi-dimensional construct [33,34,35,36,37,39]. Professionalism can be assessed as a comprehensive construct or as a facet for competency, using a variety of tools [34]. We did not intend to assess professionalism in the present study. Instead, we used PCI to assess the role models between the two hospitals and the results support that C Hospital is a better learning environment than B Hospital in cultivating professionalism for the fifth-year medical students [33]. Additionally, the present study didn’t measure psychological conditions of students to identify whether the integrated clinical clerkship program was beneficial for the mental health of the participants, which can be added as a perspective in future research directions.

## 5. Conclusions

In this study, when compared with their prospectively matched peers receiving conventional clerkships at a tertiary teaching hospital, students receiving integrated clerkships in medicine and surgery at a cancer center had a stronger sense of ethics, patient–doctor relationships, and professionalism, and in OSCE clinical skills were not inferior in performance. Our study provides an example of an effective integrated clerkship program for teaching medical humanities as well as a useful assessment framework for investigating the comparative effectiveness of clinical educational programs. The findings of this study can be used as a reference for the continuous promotion of clinical teaching of medical humanities and the rigorous evaluation of teaching effectiveness with multiple subjective and objective indicators.

## Figures and Tables

**Figure 1 ijerph-19-01882-f001:**
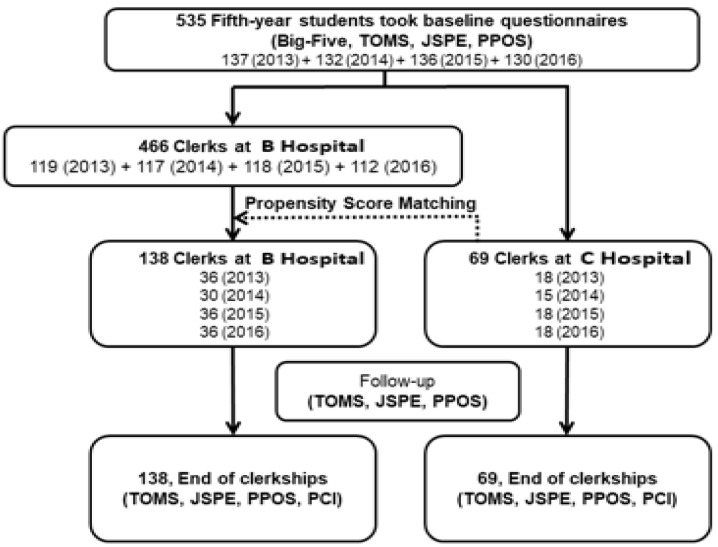
Enrollment and follow-up of medical students participating in the study for four consecutive academic years (2013–2016). Note: Big-Five, International English Big-Five Mini-Markers; JSPE, Jefferson Scale of Physician Empathy Student Version; OBS/GYN, Obstetrics/Gynecology; PCI, Professionalism Climate in Clinical Teaching Environment; PPOS, Patient–Practitioner Orientation Scale; TOMS, Task of Medicine Scale.

**Figure 2 ijerph-19-01882-f002:**
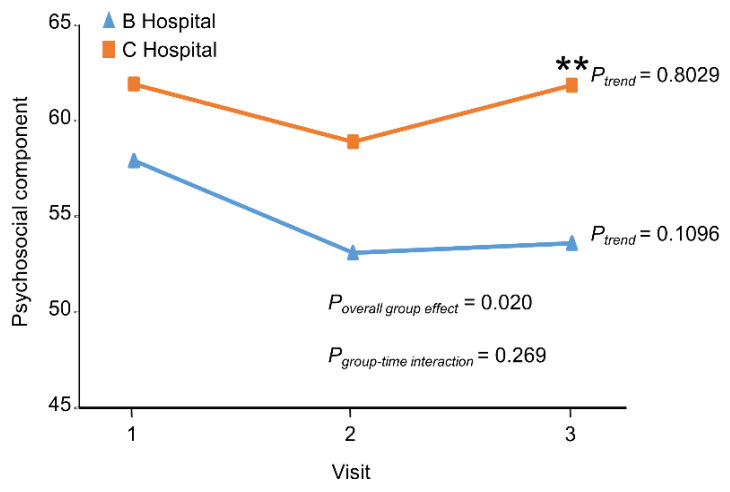
The comparison of Task of Medicine Scale (TOMS) between the B Hospital and C Hospital at baseline, at the completion of 3-month clerkship in medicine or surgery, and at the end of the 9-month clerkship program. **: *p* value for within visit mean difference <0.01.

**Figure 3 ijerph-19-01882-f003:**
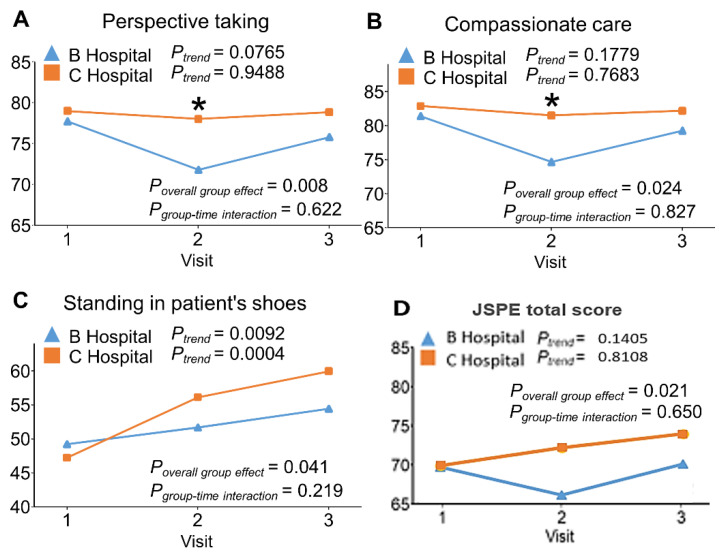
The comparison of Jefferson Scale of Physician Empathy Student Version (JSPE) between the B Hospital and C Hospital at baseline, at the completion of the 3-month clerkship in medicine or surgery, and at the end of the 9-month clerkship program. Panel (**A**). Perspective taking; Panel (**B**). Compassionate care; Panel (**C**). Standing in patient;s shoes; Panel (**D**). JSPE total score.

**Figure 4 ijerph-19-01882-f004:**
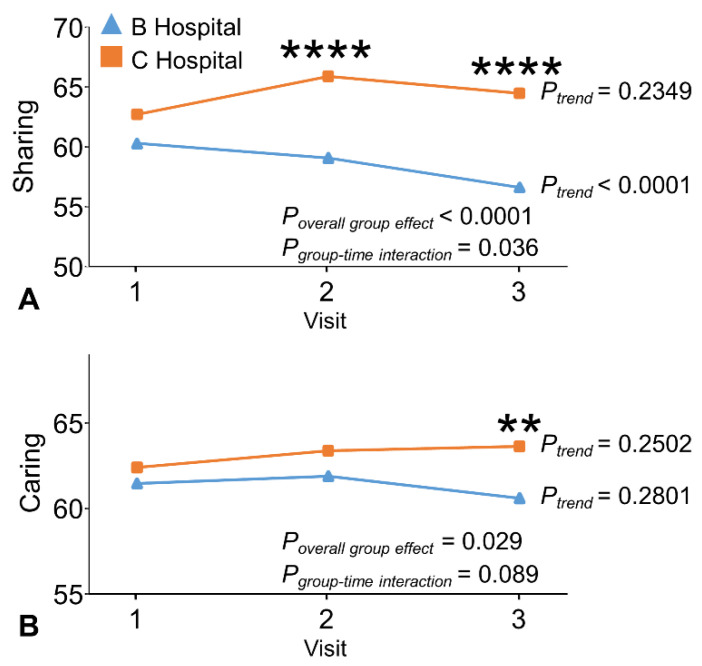
The comparison of (**A**) Sharing and (**B**) Caring components of Patient–Practitioner Orientation Scale (PPOS) between the B Hospital and C Hospital at baseline, at the completion of 3-month clerkship in medicine or surgery, and at the end of the 9-month clerkship program. ** = *p* for within visit mean difference <0.01; **** = *p* for within visit mean difference <0.0001.

**Table 1 ijerph-19-01882-t001:** Baseline characteristics of the C Hospital clerks, and comparison students (from B Hospital) prospectively selected by propensity score matching.

Variable	Overall *N* = 207	C Hospital *N* = 69	B Hospital *N* = 138	*p*-Value
Age	22.6 ± 1.1	22.7 ± 1.5	22.5 ± 0.9	0.2298
Sex				0.4912
Male	100 (48%)	31 (45%)	69 (50%)	
Female	107 (52%)	38 (55%)	69 (50%)	
Admission route				0.6898
Interview	105 (51%)	37 (54%)	68 (48%)	
Recommendation	28 (14%)	7 (10%)	21 (15%)	
Examination	73 (35%)	24 (35%)	49 (36%)	
Study Loan				0.4624
No	149 (72%)	50 (72%)	99 (72%)	
Yes	58 (28%)	19 (28%)	39 (28%)	
Part-time job				0.6580
No	126 (61%)	41 (60%)	85 (62%)	
Yes	80 (39%)	27 (40%)	53 (38%)	
Personalities ^1,2^				
Extraversion ^1,2^	40.9 ± 12.0	39.9 ± 12.4	41.4 ± 11.8	0.3978
Agreeableness ^1,2^	52.6 ± 7.5	52.5 ± 7.5	52.6 ± 7.5	0.9116
Conscientiousness ^1,2^	34.1 ± 10.2	34.3 ± 10.4	34.0 ± 10.2	0.8783
Neuroticism ^1,2^	47.7 ± 7.9	47.5 ± 8.4	47.9 ± 7.6	0.7552
Openness ^1,2^	48.6 ± 8.0	48.4 ± 8.0	48.7 ± 8.0	0.8301

^1^ All scores were linearly transformed to a 0–100 scale; ^2^ Big-Five, International English Big-Five Mini-Markers; JSPE, Jefferson Scale of Physician Empathy Student Version; PPOS, Patient–Practitioner Orientation Scale; TOMS, Task of Medicine Scale.

**Table 2 ijerph-19-01882-t002:** Comparison of humanity outcome between 2 hospitals at baseline and at the end of the clinical clerkship program.

At Baseline	Overall N = 207	C Hospital N = 69	B Hospital N = 138	*p*-Value	Non-Inferiority Test
**At baseline ^1,2^**					
**TOMS**					
Psychosocial	59.3 ± 18.3	61.9 ± 17.7	57.9 ± 8.5	0.1439	
**JSPE**					
Perspective taking	78.1 ± 11.1	79.0 ± 11.9	77.7 ± 10.7	0.4413	
Compassionate care	81.9 ± 11.8	82.9 ± 11.4	81.4 ± 12.0	0.4000	
Standing in patient’s shoes	48.6 ± 24.8	47.2 ± 26.1	49.2 ± 24.2	0.5873	
**PPOS**					
Sharing	61.1 ± 10.0	62.7 ± 10.1	60.3 ± 9.8	0.1000	
Caring	61.8 ± 8.1	62.4 ± 7.7	61.5 ± 8.2	0.4368	
**At the end of programs ^1,2^**					
**TOMS**					
Psychosocial	56.4 ± 21.0	61.9 ± 18.1	53.6 ± 21.9	0.0091	<0.0001
**JSPE**					
Perspective taking	76.8 ± 12.6	78.8 ± 15.5	75.8 ± 10.8	0.1483	0.0001
Compassionate care	80.2 ± 15.1	82.2 ± 19.1	79.3 ± 12.6	0.2558	0.0201
Standing in patient’s shoes	56.3 ± 24.7	59.9 ± 26.8	54.4 ± 23.5	0.1333	<0.0001
Combined *	69.5 ± 11.5	69.7 ± 11.9	69.4 ± 11.3	0.8840	0.0010
**PPOS**					
Sharing	59.3 ± 11.1	64.5 ± 11.0	56.6 ± 10.3	<0.0001	
Caring	61.6 ± 7.5	63.6 ± 7.5	60.6 ± 7.4	0.0063	<0.0001
**PCI**					
Peers	60.8 ± 9.3	62.7 ± 10.5	59.8 ± 8.6	0.0490	<0.0001
Residents	58.3 ± 10.5	64.2 ± 11.2	55.4 ± 8.8	<0.0001	
Attending	60.0 ± 10.8	65.7 ± 11.3	57.1 ± 9.4	<0.0001	
Faculty	69.6 ± 16.0	77.0 ± 14.9	65.8 ± 15.2	<0.0001	
**OSCE**					
Medicine	87.2 ± 6.5	87.2 ± 5.8	87.2 ± 6.9	0.9761	0.0010
Surgery	89.7 ± 3.3	90.0 ± 3.0	89.6 ± 3.4	0.3181	0.0012
Combined *	87.4 ± 5.1	87.4 ± 5.4	87.4 ± 5.0	0.9830	0.0421
6th year Internship grade (N = 152)	91.1 ± 1.9	91.0 ± 2.2	91.1 ± 1.7	0.7762	0.0529
7th year Internship grade (N = 97)	92.9 ± 1.4	93.0 ± 1.4	92.9 ± 1.3	0.6464	0.0296

^1^ A combination of obstetrics/gynecology, pediatrics, and radiology; ^2^ All scores and grades were linearly transformed to a 0–100 scale; JSPE, Jefferson Scale of Physician Empathy Student Version; OBS/GYN, Obstetrics/Gynecology; PCI, Professionalism Climate in Clinical Teaching Environment; PPOS, Patient–Practitioner Orientation Scale; TOMS, Task of Medicine Scale; *: combined scores of all components of different assessment tools

## Data Availability

The database of the present study can be made available on reasonable request to the corresponding author.

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
