# Peer review of "Evaluating Teaching Effectiveness of Medical Humanities in an Integrated Clerkship Program by a Novel Prospective Propensity Score Matching Framework"

_ijerph, 2022, doi:10.3390/ijerph19031882_

Round 1

Reviewer 1 Report

This research about medical teaching is forward-looking and meaningful. However, some parts still require revision. First, regarding whether past studies have conducted relevant research, please add a section introducing relevant literature. Second, most literature cited in this paper were dated, most of them being from 2015 or earlier. Please substantially update your cited literature. Third, why did this study choose to compare 69 students with 139 people of the same age? What was the mechanism for selecting participants? Please explain the reasons clearly. Fourth, the experimental duration was from 2013 to 2016. Could it be possible that newer research discoveries have been released after 2017? Why didn’t the authors conduct relevant research? Please list them as your research limitations. Fifth, this study only stated research findings. However, the authors did not present the meanings of what values the practical meaning of this study brings, either theoretically or practically. Please clearly state them in your revision.

Author Response

  1. Regarding whether past studies have conducted relevant research, please add a section introducing relevant literature.

Regarding your 1st and 2nd opinion, we have added a paragraph to the Introduction. In addition to historical and classic literature, we tried to cite recent empirical research literature, and revise the last sentence of the previous paragraph to improve the continuity of the two paragraphs. The content is as follows:

[Introduction, Page 2, Line 25-42]

…Our preliminary report of the first 3 years (2006-2007, 2007-2008, 2008-2009) results suggested that the pilot collaborative program is a successful model for clinical education in the teaching of core clinical competencies through direct patient care responsibilities, that both emphasizes essential clinical skills and provision of holistic care. [15]

The promotion of holistic medical care is a focus of Taiwan's medical education reform in recent years, [16] and it also responds to the international movement to address importance of the holistic medical environment [17-18]. In Taiwan, medical students are usually the top academic performers in high school, and the medical education system also reflects the high emphasis on medical professional knowledge and skills. [19] With the introduction of the concept of holistic medical education, "integration of human multi-faceted intelligence and life experience" has been considered to be a very important direction for developing a doctor with both medical technology and humanistic care. [17-19] However, when we reviewed previous studies which explored the integration of holistic medical education into clinical practice, research on students' performance in professional skills and humanistic care was rare. Preliminary research has shown that the acceptance of holistic care of medical students is related to their learning motivation, ethical cognition, and even mental health. [19-21] We believe that empirical evidence that evaluated teaching effectiveness of medical humanities in an integrated clerkship program in medical education has considerable importance for catalyzing holistic education paradigm.

  1. Most literature cited in this paper were dated, most of them being from 2015 or earlier. Please substantially update your cited literature.

Regarding your 1st and 2nd opinion, we have added a paragraph to the Introduction. In addition to the original references, we tried to cite recent empirical research literature, and revise to improve the continuity of the two paragraphs as follows:

[Introduction, Page 2, Line 25-42]

…Our preliminary report of the first 3 years (2006-2007, 2007-2008, 2008-2009) results suggested that the pilot collaborative program is a successful model for clinical education in the teaching of core clinical competencies through direct patient care responsibilities, that both emphasizes essential clinical skills and provision of holistic care. [15]

The promotion of holistic medical care is a focus of Taiwan's medical education reform in recent years, [16] and it also responds to the international movement to address importance of the holistic medical environment [17-18]. In Taiwan, medical students are usually the top academic performers in high school, and the medical education system also reflects the high emphasis on medical professional knowledge and skills. [19] With the introduction of the concept of holistic medical education, "integration of human multi-faceted intelligence and life experience" has been considered to be a very important direction for developing a doctor with both medical technology and humanistic care. [17-19] However, when we reviewed previous studies which explored the integration of holistic medical education into clinical practice, research on students' performance in professional skills and humanistic care was rare. Preliminary research has shown that the concept of holistic care of medical students is related to their learning motivation, ethical cognition, and even mental health. [19-21] We believe that empirical evidence that evaluated teaching effectiveness of medical humanities in an integrated clerkship program in medical education has considerable importance for catalyzing the strength of holistic education paradigm.

  1. Third, why did this study choose to compare 69 students with 139 people of the same age? What was the mechanism for selecting participants? Please explain the reasons clearly.

Regarding your question, we added the reply to the Method section:

[Methods, Page 4, Line 40-45]

…In order to maximize the number of participants, the researchers endeavored to invite all medical intern students from Hospital C during the research period. Through the method of prospective propensity score matching, students with similar basic background conditions as the students of Hospital C were selected to prospectively compare the teaching effectiveness of the integrated clinical clerkship program.…

  1. The experimental duration was from 2013 to 2016. Could it be possible that newer research discoveries have been released after 2017?

Thanks for the comment, we have added the following sentence to the study limitation:

[Discussion, Page 12, Line 14-17]

…In addition, the number of participants in this study was fulfilled because the number of samples has reached the pre-planning standard of the research plan. We have conducted extensive literature review; however, no newer research discoveries have been uncovered after 2017.

  1. 5. Why didn’t the authors conduct relevant research? Please list them as your research limitations.

Regarding your opinion, the first author of this article, Professor Chen-Huan Chen, is the Dean of the School of Medicine of Yang Ming Chiao Tung University. He has rich experience in medical education practice and leadership, and also participated in the conduct of the predecessor papers cited in this study:

Wang, Y. A., Chen, C. F., Chen, C. H., Wang, G. L., & Huang, A. T. (2016). A clinical clerkship collaborative program in Taiwan: Acquiring core clinical competencies through patient care responsibility. Journal of the Formosan Medical Association, 115(6), 418-425.

In recent years, he has also published other papers related to medical education:

Wu, S. H., Huang, C. C., Huang, S. S., Yang, Y. Y., Liu, C. W., Shulruf, B., & Chen, C. H. (2020). Effects of virtual reality training on decreasing the rates of needlestick or sharp injury in new-coming medical and nursing interns in Taiwan. Journal of educational evaluation for health professions, 17: 1.

Wu, C. H., Huang, Y. C., Chiang, D. H., Yang, Y. Y., Yang, L. Y., Kao, S. Y., Chen, C. H., & Lee, F. Y. (2020). A quality improvement pilot project of training nurses to use VR educational aids to increase oral cancer patients' pretreatment knowledge and satisfaction. European journal of oncology nursing: the official journal of European Oncology Nursing Society, 49, 101858.

Shin, J., Chia, Y. C., Heo, R., Kario, K., Turana, Y., Chen, C. H., ... & Li, Y. (2021). Current status of adherence interventions in hypertension management in Asian countries: A report from the HOPE Asia Network. The Journal of Clinical Hypertension, 23(3), 584-594.

  1. This study only stated research findings. However, the authors did not present the meanings of what values the practical meaning of this study brings, either theoretically or practically. Please clearly state them in your revision.

Based on your comment, we have revised some of the text in the Discussion as follows:

[Discussion, Page 10, Line 23-32]

Medical schools in Taiwan have been introducing humanities into medical education for decades, [34] make tremendous efforts to continually deepening the "holistic medical education". Several practical strategies have been proposed to enhancing curricula related to the humanistic aspects of medical training for medical students, including positive role modeling, establishing a humanistic learning environment, creating learning objectives directly related to psychosocial issues, and service learning.[34-37] These approaches respond to holistic education covering a wide range of philosophical orientations and pedagogical practices. [18] Broadly speaking, holistic education includes the part of intelligent/professional clinical technology, as well as the improvement of spiritual aspects such as ethics, caring, and spirituality. [18] [20]

[Discussion, Page 11, Line 19-21]

The effectiveness of humanity learning outcomes demonstrated in the present study was apparently associated with the innovative design of the integrated clerkship in which formal humanities education is emphasized, and also related to the hidden curriculum in which role modeling may play an imperative role. The proper design of the clinical education program and the teaching environment can overcome the limitations of institutional resources and is conducive to the cultivation of the medical humanity of medical trainees. This speculation was supported by the findings of PCI (Table 2), through which these medical students rated the professionalism of their peers and teachers in differential patterns between the two hospitals.

[Discussion, Page 11, Line 37-39]

The skills of building up patients-doctors relationship and empathy are indispensable in the cancer center and therefore helpful for the development of professionalism of medical students. These considerations generally respond to the spirit of holistic education that focuses on the diverse life experiences of human experience, rather than being limited to basic technical training. {Mahmoudi, 2012 #48;Yen-Ju Lin, 2019 #49} These may be the reasons that the students at the cancer center had a significantly higher rating of PPOS (Sharing), and PCI at the end of their clerkship training (Table 2).

Reviewer 2 Report

This is the study that aimed to Evaluate teaching effectiveness of Medical Humanities through direct patient care in an Integrated Clerkship Program. Although the result is also meaningful, there are several points to be addressed to improve.

Major
1 Please show the total score of JSPE. Although authors present the three categorized data of JSPE, total score is more important than categorized score.

2 How many cancer patients do the students take care in each hospital? I guess that even students in B hospital also take care of many cancer patients, and it might affects several parameters.

Author Response

  1. Please show the total score of JSPE. Although authors present the three categorized data of JSPE, total score is more important than categorized score.

Thank you for your suggestion. We have added an analysis of the JSPE total score, and the new results are listed in Table 2 and Figure 3.

[Results, Table 2, Page 8]

[Results, Figure 3, Page 8]

We also add the corresponding text of the new result to Results:

[Results, Page 9, Line 12-13]

Over the 3 time-points, the C Hospital students had a significantly higher JSPE Perspective taking score than their peers (Figure 3A; P for overall group effect = 0.008). Results were similar for the JSPE Compassionate care score (Figure 3B; P for overall group effect = 0.024) and JSPE total score (Figure 3D; P for overall group effect = 0.021). However, the JSPE Standing in patient’s shoes score significantly increased with time in both groups…

  1. How many cancer patients do the students take care in each hospital? I guess that even students in B hospital also take care of many cancer patients, and it might affect several parameters.

With reference to your opinion, we have added the following clarification to the Discussion:

[Discussion, Page 11, Line 49-53 to Page 12, Line 1-2]

…It must be noted that in the two participating hospitals, the training intensity and performance evaluation standards of students participating in clinical practice were comparable. According to the teaching hospital Accreditation evaluation provisions implemented by the Ministry of Health and Welfare (MOHW) in Taiwan, [44] the number of clinical practice caring inpatients for medical students in the fifth grade starts with 1 and the upper limit is 10 beds; the actual number of caring beds in Hospital B and C are both 3 or less.

Reviewer 3 Report

ijerph-1525807

Evaluating Teaching Effectiveness of Medical Humanities through Direct Patient Care in an Integrated Clerkship Program: Adopting a Novel Prospective Propensity Score Matching Framework

Suggested Title: Your paper title serves as the initial guide to the essence of your work so please revise your title so it includes the most important elements of your report. The current one is too long for an article. For example, what is your method for conducting this study? Please revise your title.

Abstract has reflected the  organizational structure of paper (i.e., presents problem/focus of study, research questions, participants, methodology, findings, key points from discussion of findings

Methods Section: explains how research design fits with research objectives, explains what type of  inquiry was used, provides step by step description of procedures, with corresponding headings, describes sampling strategy and participant recruitment, explains steps of data generation, collection, and data analysis, as well as rationale for each design choice, and tells reader what constitutes data.

 Results: Please address the following:

  1. Findings should respond to the purpose of the study, and
  2. Should be presented systematically, please relate to your research questions.

Discussion Section: does not repeat information already presented in paper, discusses how findings compare/contrast with what was known and/or not known in the literature, discusses limitations of study, discusses position on generalizability of results, discusses implications of findings, indicates area of future research.

Conclusion

Conclusion must be drawn based on research questions and purposes of your study. Please revise it.

References: Please make sure all citations that you cited be included in "references section"

Author Response

  1. Evaluating Teaching Effectiveness of Medical Humanities through Direct Patient Care in an Integrated Clerkship Program by: Adopting a Novel Prospective Propensity Score Matching Framework

Suggested Title: Your paper title serves as the initial guide to the essence of your work so please revise your title so it includes the most important elements of your report. The current one is too long for an article. For example, what is your method for conducting this study? Please revise your title.

Thanks for your suggestion, we have shortened the title to:

“Evaluating Teaching Effectiveness of Medical Humanities in an Integrated Clerkship Program by a Novel Prospective Propensity Score Matching Framework”

  1. Results: Please address the following:
  • Findings should respond to the purpose of the study, and
  • Should be presented systematically, please relate to your research questions.

Thanks for your suggestion. We have revised the clear statement of research purpose and made it a separate paragraph along with the research hypothesis to make the statement of results and conclusions more clearly aligned.

[Introduction, Page 2, Line 53-54 to Page 3, Line 1-4]

The main purposes of this study were: (1) to evaluate the training effect of the Integrated Clerkship Program on the medical humanistic literacy of trainee medical students; (2) to understand the comparative effectiveness between the Integrated Clerkship Program implemented in the specialized hospital and the conventional internship program of the tertiary teaching hospital, whether there are differences in clinical skill performance among participating medical students….

Because the text of the Results corresponds to the Tables and Figures, we place the textual statement of the research findings corresponding to the research purpose in the first paragraph of the Discussion to reduce the excessive repetition of the text.

  1. 3. Discussion Section: does not repeat information already presented in paper, discusses how findings compare/contrast with what was known and/or not known in the literature, discusses limitations of study, discusses position on generalizability of results, discusses implications of findings, indicates area of future research.

Thanks for your suggestion, we have added the following text to the Discussion and Conclusions:

[Discussion, Page 11, Line 43-48]

The results of this study reflect the findings of past research that holistic care education can help improve medical students' humanistic thinking, [19] [21] and present the different aspects of medical humanity with more diverse and detailed indicators. In addition, the results of this study also support the feasibility of implementing an educational program in specialized hospitals that balances professional knowledge and humanistic care.

[Discussion, Page 12, Line 28-29]

In addition to the integrated clerkship p  rogram itself, the organizational structure and atmosphere of the two groups of hospitals may also have substantial impacts on students' learning performance, limiting the comparability between these two hospitals. PSM can only control potential measured variables at the individual level.

  1. 4. Conclusion: Conclusion must be drawn based on research questions and purposes of your study. Please revise it.

Based on your comment, we have revised some of the text in the Conclusion as follows:

[Conclusion, Page 13, Line 3-8]

Our study provides an example of effective integrated clerkship program for teaching medical humanities as well as a useful assessment framework for investigating the comparative effectiveness of clinical educational programs. The findings of this study can be used as a reference for the continuous promotion of clinical teaching of medical humanities and the rigorous evaluation of teaching effectiveness with multiple subjective and objective indicators.

  1. 5. References: Please make sure all citations that you cited be included in "references section".

We have rechecked the consistency between the references cited in the paper and the following reference list items.

Round 2

Reviewer 1 Report

We have revisited the complete manuscript. Thank you very much again for your time and effort.